# General-purpose pre-trained model towards cross-domain molecule learning

## Abstract

Self-supervised pre-training on biomolecules has achieved remarkable success in various biochemical applications, such as drug discovery and protein design. However, in most approaches, the learning model is primarily constructed based on the characteristics of either small molecules or proteins, without exploring their potential binding interactions – an essential cross-domain relationship crucial for driving numerous biological processes. In this paper, inspired by the success of multimodal learning, we fill this gap by proposing a general-purpose foundation model named **BIT** (an abbreviation for **B**iomolecular **I**nteraction **T**ransformer), which is capable of encoding a range of biochemical entities, including small molecules, proteins, and protein-ligand complexes, as well as various data formats, encompassing both 2D and 3D structures, all within a shared Transformer backbone, via multiple unified self-supervised atom-level *denoising* tasks. We introduce *Mixture-of-Domain-Experts* (MoDE) to handle the biomolecules from diverse chemical domains and incorporate separate structural channels to capture positional dependencies in the molecular structures. The proposed MoDE allows BIT to enable both deep fusion and domain-specific encoding and learn cross-domain relationships on protein-ligand complexes with 3D cocrystal structures. Experimental results demonstrate that BIT achieves exceptional performance in both protein-ligand binding and molecular learning downstream tasks, including binding affinity prediction, virtual screening, and molecular property prediction.

## 1 Introduction

In the past few years, self-supervised pre-training of the foundation model has witnessed remarkable success in natural language processing (Devlin et al., 2018; Brown et al., 2020) and computer vision (Chen et al., 2020; He et al., 2022). Recently, pre-training on biomolecules has attracted growing attention. By fine-tuning the large-scale pre-trained model, one can significantly improve the performance on diverse biological downstream tasks, such as molecular property prediction (Rong et al., 2020), protein structure prediction (Lin et al., 2023) and protein design (Madani et al., 2023). Thus, substantial efforts have been devoted to biomolecule pre-training to leverage the potential inherent in the large-scale unlabeled molecule corpus, including molecular graphs and protein sequences (Hu et al., 2020; Rives et al., 2021). However, the majority of existing approaches are tailored to a single data domain, focusing on either small molecules or proteins. This restricts the pre-trained model from capturing molecular interactions across distinct chemical domains. Consequently, it limits the learning performance in the downstream tasks that highly depend on this information, such as structure-based binding affinity prediction and virtual screening (Ain et al., 2015).

The interactions between proteins and small molecules, known as ligands, play a pivotal role in orchestrating molecular-level biological processes (Tomasi & Persico, 1994). Understanding the underlying principles of these interactions is crucial in scientific fields. Besides, it is essential for a range of biomedical applications, particularly in structure-based drug discovery (SBDD) (Anderson, 2003; Vamathevan et al., 2019). To be more precise, by considering the 3D geometry of the binding pocket on the target protein, medicinal chemists strive to identify prospective drug candidates capable of modulating the function of the target. Recent studies have shown the significant potential of deep learning methods in modeling molecular interactions and facilitating the SBDD process (Li et al., 2021b; Luo et al., 2021; Corso et al., 2022). However, given the vastness of the chemical space

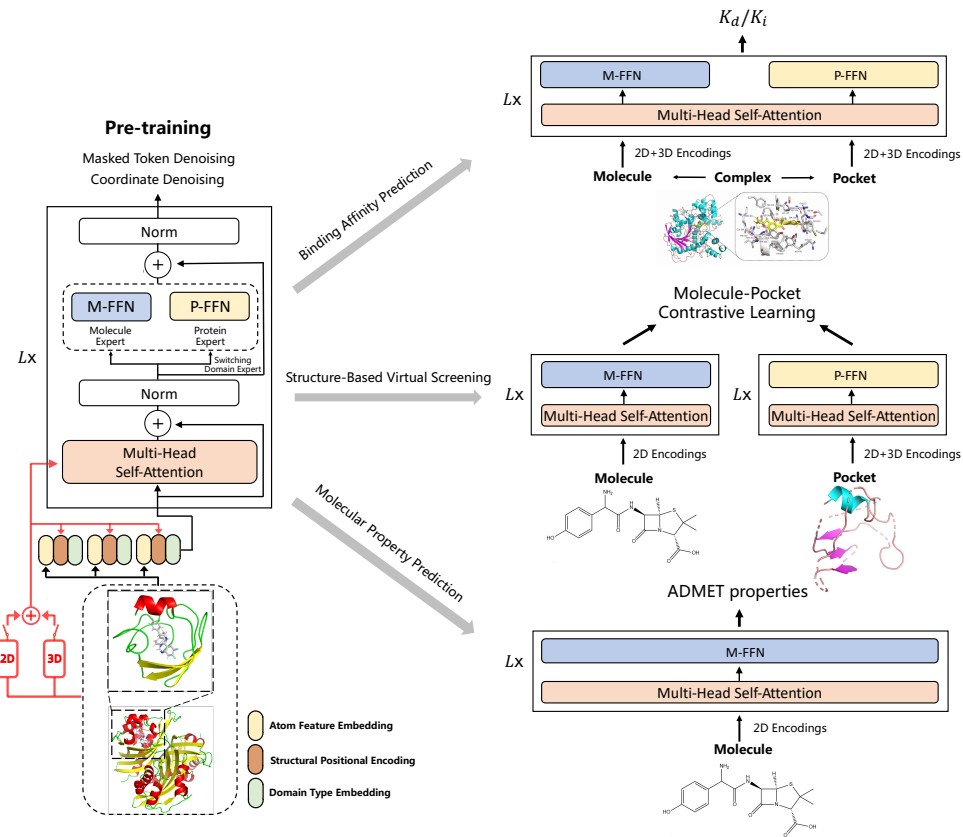

Figure 1: Overview of BIT pre-training and fine-tuning. We perform denoising tasks on protein-ligand complexes with a general-purpose foundation model named BIT. During fine-tuning, BIT can be used as (a) fusion encoder for binding affinity prediction; (b) dual encoder for virtual screening; (c) molecule encoder for molecular property prediction.

and the scarcity of task-specific labeled data, building a highly accurate prediction model for such problems remains a significant challenge.

More recently, preliminary attempts have been dedicated to pre-training a model that is explicitly designed to capture cross-domain dependencies between protein pockets and ligands, as exemplified by CoSP (Gao et al., 2022). Specifically, it distinguishes the two different domains as independent signals and adopts a dual-encoder architecture similar to CLIP (Radford et al., 2021) to encode pockets and ligands separately, then employs contrastive learning (Oord et al., 2018; Chuang et al., 2020) to learn a shared embedding space in which bindable pockets and ligands are pulled closer, while unbindable pocket-ligand pairs are pushed apart. However, the molecular interaction is merely inferred through a shallow interaction module, which involves the dot product of the pocket and ligand feature vectors, without considering inter-molecular connection information. As a result, CoSP remains unsatisfactory for addressing complex protein-ligand binding tasks. To effectively integrate both intra-molecular and inter-molecular interactions from protein-ligand complexes, a more intricate alignment between proteins and ligands is required.

Considering the aforementioned issues, it is straightforward to adopt multimodal learning (Xu et al., 2023b) to leverage all essential information available across diverse chemical domains, encompassing various data formats. The Transformer (Vaswani et al., 2017) is chosen as a preferred backbone as its variants have demonstrated effectiveness in modeling text (Devlin et al., 2018), images (Dosovitskiy et al., 2021), graphs (Rampášek et al., 2022), and molecule (Ying et al., 2021). Furthermore, it can handle multiple modalities in a unified manner. For example, Transformer-based multimodal models (Kim et al., 2021; Wang et al., 2023a) have significantly advanced vision-language understanding and generation ability, resulting in improved performance across various vision-language tasks.

In this work, we present a general-purpose pre-trained model within the *protein-ligand pre-training* paradigm. This model encodes molecules across a wide range of biochemical domains, including small molecules, proteins, and protein-ligand complexes, as well as diverse data formats, encompassing both 2D and 3D structures, all within a unified Transformer framework, referred to as **B**iomolecular **I**nteraction **T**ransformer (**BIT**). We construct our model upon Transformer-M (Luo et al., 2022), a model renowned for its flexibility and effectiveness in handling both 2D and 3D structural data. Then we enhance it to capture multi-domain specificity and inter-domain relationships by incorporating *Mixture-of-Domain-Experts* (MoDE). In each Transformer block, we replace the feed-forward network with two distinct domain experts, namely the molecule expert and the protein expert, while retaining a shared self-attention module across domains to facilitate alignment between different domains. In BIT, each input atom token is routed to its respective domain expert, enabling BIT to function as either a fusion encoder to model molecular interactions in protein-ligand complexes or as a dual encoder to separately encode small molecules and proteins.

To learn more precise cross-domain representations, we pre-train BIT on both protein-ligand complexes with 3D cocrystal structures (Wei et al., 2023) and large-scale small molecules with 3D equilibrium structures (Hu et al., 2021) in a unified manner via denoising tasks for both continuous atom coordinates and categorical atom types. Finally, we demonstrate the superiority of the proposed BIT through extensive experiments across various downstream tasks, including both protein-ligand binding and molecular learning. When employed as a fusion encoder, BIT consistently outperforms specialized baselines by a decent margin in binding affinity prediction. When used as a dual encoder, BIT still achieves state-of-the-art performance while offering significantly faster inference speed in virtual screening. Moreover, BIT surpasses related state-of-the-art pre-trained models in a series of molecular property prediction tasks.

The main contributions of this work are summarized as follows:

- We present BIT, a general-purpose foundation model designed to encode a range of biochemical entities, including small molecules, proteins, and protein-ligand complexes, across various data formats, encompassing both 2D and 3D structures, all within a unified Transformer backbone.
- We pre-train BIT on protein-ligand complexes with 3D cocrystal structures, alongside large-scale small molecule ligands with 3D equilibrium structures, to learn cross-domain biomolecule relationships using cross-domain attention.
- Experiments verify that BIT achieves exceptional performance in downstream tasks, including both protein-ligand binding and molecular learning, after fine-tuning.

## 2 RELATED WORK

### 2.1 MOLECULAR REPRESENTATION LEARNING

Learning meaningful and effective molecular representations is fundamental to AI-driven drug discovery. Self-supervised pre-training serves as a powerful tool in this area, thanks to the availability of the abundance of molecule data. Recently, several self-supervised pre-training models have been proposed separately for small molecules or proteins.

**Pre-training on small molecules:** Initially, researchers employ sequence-based pre-training strategies on string-based molecular data such as SMILES (Weininger, 1988). Representative works include SMILES-BERT (Wang et al., 2019) and ChemBERTa (Chithrananda et al., 2020). As molecular graphs can provide richer 2D structural information, more efforts (Hu et al., 2020; Rong et al., 2020; Wang et al., 2022a) have focused on pre-training graph neural networks (Xu et al., 2019) or Transformers (Vaswani et al., 2017) on molecular graphs. Moreover, there are recent studies aiming to explore pre-training on 3D molecular structures to improve model performance in predicting molecular properties using 3D structural data (Zaidi et al., 2022; Zhou et al., 2023; Feng et al., 2023).

**Pre-training on proteins:** Learning effective protein representations is also of great importance, such as for protein understanding and generation. Protein language models have achieved remarkable success in capturing biological co-evolutionary information from millions of diverse protein sequences (Elnaggar et al., 2021; Lin et al., 2023), or families of evolutionarily related sequences (Rao et al., 2021). Beyond these sequence-based approaches, there is a growing interest in exploring pre-training techniques for protein structures (Zhou et al., 2023; Zhang et al., 2023).

For more comprehensive reviews, we refer the reader to Xia et al. (2023b); Ferruz & Höcker (2022). While most prior work constructed models based on the characteristics of either small molecules or proteins, our work aims to enhance molecular representation learning by incorporating additional cross-domain relationships learned from biologically relevant protein-ligand complexes.

## 2.2 MULTIMODAL REPRESENTATION LEARNING

In recent years, multimodal representation learning has gained significant attention and has been extensively studied to enhance understanding across various areas, including image analysis (Radford et al., 2021), video processing (Sun et al., 2019), and speech recognition (Ao et al., 2022), often by incorporating additional textual information. Among these multimodal learning applications, Transformer (Vaswani et al., 2017; Dosovitskiy et al., 2021) has become a critical building block, thanks to its outstanding performance in learning from the corresponding monomodal data and its flexibility in aligning and integrating information across multimodal data sources. There are three main types of architectures to cater to different multimodal learning requirements: dual encoder (Radford et al., 2021; Jia et al., 2021) for efficient retrieval, fusion encoder (Kim et al., 2021; Li et al., 2021a) for deep understanding, and encoder-decoder architectures (Wang et al., 2022b) for generation. Some research (Li et al., 2022; Bao et al., 2022; Wang et al., 2023a) have explored effective ways to integrate the strengths of these architectures. Recently, multimodal learning has also found applications in the biomedical field. There have been early attempts to enhance molecular representation learning by leveraging the correspondence and consistency between 2D topological structures and 3D geometric views (Liu et al., 2022; Stärk et al., 2022; Liu et al., 2023) or incorporating biomedical text (Liu et al., 2022; Xu et al., 2023a).

## 2.3 STRUCTURE-BASED DRUG DISCOVERY

Structure-based drug discovery (SBDD) refers to a systematic scientific approach to design and develop new drugs by leveraging the detailed physical structure of the binding protein or molecular target. It involves analyzing the structure of the target, understanding its function, and designing molecules capable of interacting with the target in a specific and favorable manner to regulate its activity. To complement labor-intensive traditional methods, geometric deep learning algorithms (Atz et al., 2021) have recently been proposed to improve the efficiency and performance of various stages in the SBDD process, including binding site identification (Sverrisson et al., 2021), affinity prediction (Li et al., 2021b), virtual screening (Torng & Altman, 2019), *de novo* molecule generation (Luo et al., 2021), etc. Our proposed versatile model will further streamline and enhance this process.

# 3 METHODS

In this section, we present the details of BIT, a general-purpose pre-trained model designed to encode molecules across various biochemical domains, including small molecules, proteins, and protein-ligand complexes, in different data formats, including 2D and 3D structures. BIT can be fine-tuned as either a fusion encoder to model intricate molecular interactions within protein-ligand complexes for precise binding affinity prediction, or as a dual encoder to enable efficient virtual screening.

## 3.1 INPUT REPRESENTATIONS

In biochemical applications, data are collected in the form of molecules represented at different levels of granularity, such as atoms, residues, and nucleobases. However, all molecules can be uniformly represented as sets of atoms held together by attractive or repulsive forces. To more effectively capture and transfer atom-level knowledge across different domains, we propose to share atom embeddings and incorporate domain embeddings to distinguish between small molecules and proteins.

Both small molecule, denoted as $\mathcal{M}$, and protein, denoted as $\mathcal{P}$, can be represented as a geometric graphs of atoms $\mathcal{G} = (\mathcal{V}, \mathcal{E})$. Here $\mathcal{V} = (\boldsymbol{X}, \vec{R})$ includes all atoms and $\mathcal{E}$ includes all chemical bonds. In a molecule consisting of $n$ atoms, $\boldsymbol{X} \in \mathbb{R}^{n \times d}$ denotes a set of atom feature vectors, $\vec{R} \in \mathbb{R}^{n \times 3}$ denotes a set of atom Cartesian coordinates, and $e_{ij} \in \mathcal{E}$ denotes the feature vector of the edge between atoms $i$ and $j$ if the edge exists. The molecule and protein input representations are computed via summing atom feature embeddings $\boldsymbol{X}$, structural positional encodings $\Psi \in \mathbb{R}^{n \times d}$ (Ying et al.,

2021; Shi et al., 2022), and the corresponding domain-type embedding vectors $\boldsymbol{m}_{\text{type}}, \boldsymbol{p}_{\text{type}} \in \mathbb{R}^d$. Following Ying et al. (2021), we introduce special virtual nodes [M_VNode] for small molecules and [P_VNode] for proteins, and make connection between virtual node and each atom node individually.

It is noteworthy that we only use the binding pocket as the model input rather than the entire protein primarily for the following two reasons: (1) the binding pocket is the paramount region of protein-ligand interaction, experiencing the most significant spatial alterations during the binding process and providing sufficient insight into molecular interactions; (2) the binding pocket contains significantly fewer atoms than the entire protein, leading to lower computational costs and faster training speeds.

Given a protein-ligand complex $< \mathcal{M}, \mathcal{P} >$ with 3D cocrystal structures, we first identify the binding pocket as the protein atoms located within a minimum distance of 5 Å from the ligand, as suggested in Muegge & Martin (1999). Then we input the extracted pocket-ligand complex into BIT to learn contextualized representations.

## 3.2 BACKBONE

Recently, several studies have extended the Transformers to model molecules (Rong et al., 2020; Ying et al., 2021; Rampášek et al., 2022; Luo et al., 2022). The vanilla Transformer architecture comprises stacked Transformer blocks (Vaswani et al., 2017). Each Transformer block consists of two components: a multi-head self-attention (MSA) layer followed by a feed-forward network (FFN). Layer normalization (LN) (Ba et al., 2016) is applied after both the MSA and FFN. Let $\boldsymbol{H}_{l-1}$ denotes the input, the $l$-th Transformer block works as follows:

$$\boldsymbol{H}'_l = \text{LN}(\text{MSA}(\boldsymbol{H}_{l-1}) + \boldsymbol{H}_{l-1}) \tag{1}$$

$$\boldsymbol{H}_l = \text{LN}(\text{FFN}(\boldsymbol{H}'_l) + \boldsymbol{H}'_l) \tag{2}$$

For our general-purpose modeling, we start with Transformer-M (Luo et al., 2022), a model known for its versatility and effectiveness in handling both 2D or 3D molecule data. To provide a comprehensive overview, we briefly introduce the core concept of Transformer-M here and recommend that readers refer to Luo et al. (2022) for more technical details. Transformer-M introduces two separate channels to encode 2D and 3D structural information and integrate them into the MSA module as bias terms. The modified attention matrix $\boldsymbol{A}$ is calculated as:

$$\boldsymbol{A}(\boldsymbol{H}) = \text{softmax}\left( \frac{\boldsymbol{H}\boldsymbol{W}_Q(\boldsymbol{H}\boldsymbol{W}_K)^\top}{\sqrt{d_K}} + \underbrace{\Phi^{\text{SPD}} + \Phi^{\text{Edge}}}_{\text{2D pair-wise channel}} + \underbrace{\Phi^{\text{3D Distance}}}_{\text{3D pair-wise channel}} \right) \tag{3}$$

where $\boldsymbol{W}_Q, \boldsymbol{W}_K \in \mathbb{R}^{d \times d_K}$ are learnable weight matrices, the 2D terms ($\Phi^{\text{SPD}}$ and $\Phi^{\text{Edge}}$) and the 3D term ($\Phi^{\text{3D Distance}}$) originate from Ying et al. (2021) and Shi et al. (2022) respectively. To simplify the illustration, we omit the attention head index $h$ and layer index $l$. When molecules are associated with specific 2D or 3D structural information, the corresponding channel will be activated, while the other will be disabled. In combination with the dropout-like 2D-3D joint training strategy (Luo et al., 2022), where the format of structural information for each data instance is randomly selected, Transformer-M learns to identify chemical knowledge from different data formats and generates meaningful semantic representations for each data format.

To further encode molecules across biochemical domains and learn cross-domain molecular representations enriched with molecular interaction knowledge, we propose to extend Transformer-M with a Mixture-of-Domain-Experts (MoDE) mechanism, employing specialized expert networks for different domains. As shown in Figure 1, each Transformer block in BIT consists of a shared MSA module and two FFNs, presenting domain experts, namely the molecule expert and the protein expert. In contrast to conventional mixture-of-experts layer (Shazeer et al., 2017; Fedus et al., 2022), which routes input tokens by a trainable gating network, we directly assign an expert to process each atom token based on its molecule data domain. Sharing the MSA module encourages the model to align protein and ligand, while employing MoDE in place of the FFN encourages the model to capture domain-specific knowledge. The Transformer block of BIT can be abstractly summarized as follows:

$$\boldsymbol{H}'_l = \text{LN}(\text{MSA-M}(\boldsymbol{H}_{l-1}) + \boldsymbol{H}_{l-1}) \tag{4}$$

$$\boldsymbol{H}_l = \text{LN}(\text{MoDE-FFN}(\boldsymbol{H}'_l) + \boldsymbol{H}'_l) \tag{5}$$

where MSA-M denotes the variant of MSA used in Transformer-M.

Thanks to MoDE, BIT decouples the encoding process across different domains. As a result, BIT can be fine-tuned to function as either a fusion encoder or a dual encoder, depending on the specific formulation of various downstream protein-ligand binding tasks. Further discussion on this aspect is presented in Section 3.4.

**Remarks.** Our proposed MoDE seamlessly integrates with Transformer-M. Both of them employ shared self-attention modules for unified modeling and use distinct parameters to capture the data specificity among different inputs. The main difference is that MoDE captures instance-level domain specificity with separate expert networks, while Transformer-M locates pairwise structural specificity in the MSA via separate bias terms. In summary, our extension is straightforward yet highly effective.

## 3.3 PRE-TRAINING BIT

We pre-train BIT on both protein-ligand complex and small molecule datasets. We use the Q-BioLiP database (Wei et al., 2023) as the complex corpus. It contains approximately 1.0 million biologically relevant interactions associated with 3D cocrystal structures. This dataset is sourced from the Protein Data Bank (Berman et al., 2000) through manual process (Yang et al., 2012). To prevent potential overfitting to a limited portion of the chemical space represented by ligands in the Q-BioLiP dataset, we additionally pre-train BIT on large-scale small-molecule-only corpus. For this purpose, we incorporate the PCQM4Mv2 dataset (Nakata & Shimazaki, 2017), which has been widely used for 3D molecular pre-training (Zaidi et al., 2022; Wang et al., 2023b).

To ensure the scalability of the pre-training process, we employ a unified corrupt-then-recover objective to pre-train BIT. During this pre-training approach, we randomly corrupt the continuous atom coordinates and the categorical atom types of single-domain molecules and ligands from protein-ligand complexes, and guide BIT to restore the original states. The detailed explanations of the two denoising tasks are provided below.

**Coordinate denoising** aims to learn meaningful representations that capture the inter-atomic interactions within the molecular structure. It has been demonstrated to be effective in improving 3D molecular property prediction (Zaidi et al., 2022). Theoretically, this objective can be interpreted as learning an approximate molecular force field from equilibrium structures (Zaidi et al., 2022). Thus, we can extend coordinate denoising to protein-ligand complexes, as the experimentally-determined cocrystal structures of the complexes typically represent equilibrium conformations and correspond to local energy minima. To further capture the inter-molecular interactions, we encourage the model to restore the corrupted ligand pose based on the information from both the ligand and pocket.

Formally, let $\vec{R} = \{\vec{r}_1, \vec{r}_2, ..., \vec{r}_n\}, \vec{r}_i \in \mathbb{R}^3$ denote the binding pose of a bound ligand. We perturb it by adding independent and identically distributed ($i.i.d.$) Gaussian noise to its atomic coordinates $\vec{r}_i$. The resulting noisy atom positions are denoted as $\hat{R} = \{\vec{r}_1 + \sigma\vec{\epsilon}_1, \vec{r}_2 + \sigma\vec{\epsilon}_2, ..., \vec{r}_n + \sigma\vec{\epsilon}_n\}$, where $\vec{\epsilon}_i \sim \mathcal{N}(\vec{0}, \boldsymbol{I})$ and $\sigma$ is a hyperparameter controlling the noise scale. The model is trained to predict the noise from the noisy input. The output of the last Transformer block is then fed into an SE(3) equivariant prediction head (Shi et al., 2022), driven by the denoising loss $\mathcal{L}_{pos} = \frac{1}{|\mathcal{V}|}\sum_{i \in V} \|\hat{\vec{\epsilon}}_i - \vec{\epsilon}_i\|^2$.

**Masked token denoising** aims to learn fundamental physicochemical information contained within the molecules or complexes by modeling the dependency between their atoms. This task is similar to the masked language modeling (MLM) task used in BERT (Devlin et al., 2018) and has achieved remarkable performance in molecular pre-training (Hu et al., 2020). As discussed in Austin et al. (2021), MLM can be interpreted as a categorical denoising process. Given an input molecule, we randomly mask 15% of its atoms and predict each masked atom based on its contextualized representation extracted by BIT. The cross-entropy prediction loss for this task is denoted as $\mathcal{L}_{atom}$.

## 3.4 FINE-TUNING BIT ON DOWNSTREAM TASKS

As BIT is designed to be a general-purpose cross-domain pre-train model, it is straightforward to supervised fine-tune it with task-specific data to adapt to various protein-ligand binding tasks.

**Protein-ligand binding affinity prediction.** As aforementioned, our model can serve as a fusion encoder to model the molecular interactions between proteins and ligands. Therefore, we extract the

final encoding vector from the special token [M_VNode] as the representation of the protein-ligand complexes and feed it to a task-specific prediction head to make the final prediction.

**Structure-based virtual screening.** We formulate large-scale virtual screening as a pocket-to-ligand retrieval task. In this task, our model is used as a dual encoder to encode both 3D protein pockets and 2D ligands to vectors of equal length. In fine-tuning, the pre-trained model is further optimized on task-specific data using contrastive learning. During inference, we compute representations of the target pocket and all candidate ligands, and then obtain pocket-to-ligand similarity scores of all possible pocket-ligand pairs using dot products. Hits are identified as ligands that exhibit a high level of similarity to the target pocket. This approach allows for much faster inference speeds than fusion encoder-based methods, which require preliminary molecular docking.

## 4 EXPERIMENTS

In this section, we pre-train BIT and extensively evaluate BIT on well-established public benchmarks, including both protein-ligand binding tasks and molecular learning tasks.

### 4.1 PRE-TRAINING SETUPS

#### 4.1.1 DATASETS

We pre-train BIT on both protein-ligand complex data and small molecule data. For complex data, we use the Q-BioLiP database (Wei et al., 2023), which contains 967,085 biological relevant interactions associated with 3D cocrystal structures as of June 14th, 2023. Q-BioLiP is an updated version of the original BioLiP database (Yang et al., 2012), where protein-ligand interactions are based on the quaternary structure rather than the single-chain monomer structure. This alteration provides higher-quality interactions for analyzing the binding mode. Since our primary focus is on regular ligands, i.e., small molecules, we filter out complexes containing metal ions and DNA/RNA ligands. For small molecule data, we utilize the PCQM4Mv2 dataset from the OGB Large-Scale Challenge (Hu et al., 2021), which has 3.4M organic molecules. These molecules are characterized by their 3D structures at equilibrium, calculated using density functional theory (DFT).

#### 4.1.2 TRAINING SETTINGS

Our model adopts the same network configuration as Transformer-M (Luo et al., 2022). We employ a 12-layer Transformer with a hidden size of 768 and 32 attention heads. We use AdamW optimizer (Loshchilov & Hutter, 2018) with hyper-parameter $\epsilon$ set to 1e-8 and $(\beta_1, \beta_2)$ set to (0.9,0.999). The gradient clip norm is set to 5. The peak learning rate is set to 2e-4, and we employ a 12k-step warm-up stage followed by a linear decay scheduler. The total training steps are 200k. Each batch contains 1024 samples, including 512 small molecules and 512 pocket-ligand complexes. We adopt the 2D-3D joint training strategy proposed in (Luo et al., 2022). In the coordinate denoising objective, $\sigma$ is set to 0.2. All models are trained on 64 NVIDIA Tesla V100 GPUs for approximately 2 days.

### 4.2 PROTEIN-LIGAND BINDING TASKS

#### 4.2.1 BINDING AFFINITY PREDICTION

In this task, the pre-trained model is fine-tuned to predict binding affinities $pK_a$ (or $-\log K_d$, $-\log K_i$) for protein-ligand complexes. Following previous studies (Li et al., 2021b; Luo et al., 2022), we perform fine-tuning experiments using the PDBbind v2016 dataset (Wang et al., 2004; 2005). The PDBbind dataset consists of three subsets: the general set, which includes 13,283 protein-ligand complexes; the refined set, comprising 4,057 complexes selected from the general set for higher data quality, and the core set, consisting of 285 complexes chosen for the highest data quality (Su et al., 2018). We fine-tune the pre-trained BIT using the refined set. To prevent data leakage, we remove the data instances in the core set from the refined set. We evaluate the prediction performance using metrics such as Pearson's correlation coefficient (R), Mean Absolute Error (MAE), Root-Mean Squared Error (RMSE), and Standard Deviation (SD) (Su et al., 2018).

We compare BIT with DMPNN (Yang et al., 2019), MAT (Maziarka et al., 2020), DimeNet (Gasteiger et al., 2020), CMPNN (Song et al., 2020), SIGN (Li et al., 2021b), MBP (Yan et al., 2023), and

Table 1: Binding affinity prediction results on the PDBbind core set.

| Method | R ↑ | MAE ↓ | RMSE ↓ | SD ↓ |
|---|---|---|---|---|
| DMPNN | $0.729_{\pm0.006}$ | $1.188_{\pm0.009}$ | $1.493_{\pm0.016}$ | $1.489_{\pm0.014}$ |
| MAT | $0.747_{\pm0.013}$ | $1.154_{\pm0.037}$ | $1.457_{\pm0.037}$ | $1.445_{\pm0.033}$ |
| DimeNet | $0.752_{\pm0.010}$ | $1.138_{\pm0.026}$ | $1.453_{\pm0.027}$ | $1.434_{\pm0.023}$ |
| CMPNN | $0.765_{\pm0.009}$ | $1.117_{\pm0.031}$ | $1.408_{\pm0.028}$ | $1.399_{\pm0.025}$ |
| SIGN | $0.797_{\pm0.012}$ | $1.027_{\pm0.025}$ | $1.316_{\pm0.031}$ | $1.312_{\pm0.035}$ |
| MBP | $0.825_{\pm0.008}$ | $0.999_{\pm0.024}$ | $1.263_{\pm0.023}$ | $1.229_{\pm0.026}$ |
| Transformer-M | $0.830_{\pm0.011}$ | $0.940_{\pm0.006}$ | $1.232_{\pm0.013}$ | $1.207_{\pm0.007}$ |
| BIT | $\mathbf{0.842}_{\pm0.002}$ | $\mathbf{0.927}_{\pm0.008}$ | $\mathbf{1.179}_{\pm0.007}$ | $\mathbf{1.173}_{\pm0.006}$ |

Table 2: Virtual screening results on the DUD-E dataset.

| Method | AUC ↑ | $RE_{0.5\%}$ ↑ | $RE_{1.0\%}$ ↑ | $RE_{2.0\%}$ ↑ | $RE_{5.0\%}$ ↑ |
|---|---|---|---|---|---|
| NNScore | 58.4 | 4.17 | 2.98 | 2.46 | 1.89 |
| RF-Score | 62.2 | 5.63 | 4.27 | 3.50 | 2.68 |
| Vina | 71.6 | 9.14 | 7.32 | 5.88 | 4.44 |
| 3DCNN | 86.8 | 42.56 | 29.65 | 19.36 | 10.71 |
| Graph CNN | 88.6 | 44.41 | 29.75 | 19.41 | 10.74 |
| CoSP | 90.1 | 51.05 | 35.98 | 23.68 | 12.21 |
| DrugVQA | 97.2 | 88.17 | 58.71 | 35.06 | 17.39 |
| BIT | **98.4** | **141.63** | **78.31** | **42.43** | **18.44** |

Transformer-M (Luo et al., 2022). We report the official results of baselines from Li et al. (2021b); Luo et al. (2022); Yan et al. (2023). As presented in Table 1, BIT consistently outperforms pre-training baselines and other approaches tailored for binding affinity prediction across all evaluation metrics, demonstrating the effectiveness of BIT in capturing intricate molecular interactions present in complexes.

### 4.2.2 STRUCTURE-BASED VIRTUAL SCREENING

Structure-based virtual screening of potential drug-like molecules against a protein target of interest, as outlined by Lionta et al. (Lionta et al., 2014) is a critical goal in structure-based drug discovery. This task is to identify the molecules with the highest likelihood of binding to protein pockets with known 3D structures. We choose the widely-used DUD-E dataset (Mysinger et al., 2012) for our model evaluation, following previous study (Gao et al., 2022). The DUD-E dataset comprises 102 targets across different protein families. Each target, on average, is assigned 224 binding compounds and over 10,000 decoys. These decoys are physically similar to the active compounds but differ in terms of their topology. We adopt a four-fold cross-validation strategy and use the same data split approach outlined in GraphCNN (Torng & Altman, 2019). In our data splits, we ensure that no two folds contain targets with greater than 75% sequence identity. We provide results in terms of the AUC-ROC and ROC enrichment (RE) scores. The RE score measures early enrichment and is calculated as the ratio of the true positive rate (TPR) to the false positive rate (FPR) at a given FPR threshold. Here, we report the RE scores at 0.5%, 1.0%, 2.0%, and 5.0% FPR thresholds.

Since most of the protein-ligand pairs of interest do not have experimentally solved cocrystal structures, conventional affinity prediction models that rely on this information must be complemented with molecular docking software, such as AutoDock (Trott & Olson, 2010). However, this integration often leads to significant computational expenses, particularly in large-scale virtual screening tasks. By framing virtual screening as a pocket-to-ligand retrieval task, BIT can be adopted as a dual encoder. We encode 3D protein pockets and 2D molecular graphs separately to obtain their representations in a shared subspace and compute their similarity scores by the dot product. During fine-tuning, BIT is optimized using the contrastive loss function InfoNCE (Oord et al., 2018), with 64 randomly sampled decoys per active compound.

In this task, we compare BIT with NNScore (Durrant & McCammon, 2010), RF-Score (Ballester & Mitchell, 2010), Vina (Trott & Olson, 2010), 3DCNN (Ragoza et al., 2017), Graph CNN (Torng & Altman, 2019), CoSP (Gao et al., 2022), and DrugVQA (Zheng et al., 2020). As presented in Table 2, BIT achieves superior performance compared to the baselines. Furthermore, BIT is not required to jointly encode all potential pocket-ligand pairs and can store the pre-computed representations of pockets and ligands. This enables it to achieve high screening efficiency without compromising learning precision. In the empirical study, we can compute representations for one billion molecules in an ultra-large-scale screening library (e.g., ZINC (Irwin & Shoichet, 2005)) in just under 2 days using a single NVIDIA V100 GPU.

### 4.3 MOLECULAR PROPERTY PREDICTION

In addition to the protein-ligand binding task, we also assess the capabilities of BIT in the molecular property prediction task, where BIT is used as an encoder for small molecules. In this task, we aim to predict the absorption, distribution, metabolism, excretion, and toxicity properties of molecules. We consider eight binary classification datasets from the MoleculeNet benchmark (Wu et al., 2018).

Table 3: Molecular property prediction results (with 2D topology only) on the MoleculeNet benchmark. The best and second best results are marked bold and bold, respectively.

| Methods | BBBP ↑ | Tox21 ↑ | ToxCast ↑ | SIDER ↑ | ClinTox ↑ | MUV ↑ | HIV ↑ | BACE ↑ | Avg ↑ |
|---|---|---|---|---|---|---|---|---|---|
| AttrMask (Hu et al., 2020) | 65.0±2.36 | 74.8±0.25 | 62.9±0.11 | 61.2±0.12 | **87.7±1.19** | 73.4±2.02 | 76.8±0.53 | 79.7±0.33 | 72.68 |
| ContextPred (Hu et al., 2020) | 65.7±0.62 | 74.2±0.06 | 62.5±0.31 | 62.2±0.59 | 77.2±0.88 | 75.3±1.57 | 77.1±0.86 | 76.0±2.08 | 71.28 |
| GraphCL (You et al., 2020) | 69.7±0.67 | 73.9±0.66 | 62.4±0.57 | 60.5±0.88 | 76.0±2.65 | 69.8±2.66 | 78.5±1.22 | 75.4±1.44 | 70.78 |
| InfoGraph (Sun et al., 2020) | 67.5±0.11 | 73.2±0.43 | 63.7±0.50 | 59.9±0.30 | 76.5±1.07 | 74.1±0.74 | 75.1±0.99 | 77.8±0.88 | 70.96 |
| GROVER (Rong et al., 2020) | 70.0±0.10 | 74.3±0.10 | 65.4±0.40 | 64.8±0.60 | 81.2±3.00 | 67.3±1.80 | 62.5±0.90 | 82.6±0.70 | 71.01 |
| MolCLR (Wang et al., 2022a) | 66.6±1.89 | 73.0±0.16 | 62.9±0.38 | 57.5±1.77 | 86.1±0.95 | 72.5±2.38 | 76.2±1.51 | 71.5±3.17 | 70.79 |
| GraphMAE (Hou et al., 2022) | 72.0±0.60 | 75.5±0.60 | 64.1±0.30 | 60.3±1.10 | 82.3±1.20 | 76.3±2.40 | 77.2±1.00 | 83.1±0.90 | 73.85 |
| Mole-BERT (Xia et al., 2023a) | 71.9±1.60 | 76.8±0.50 | 64.3±0.20 | 62.8±1.10 | 78.9±3.00 | 78.6±1.80 | 78.2±0.80 | 80.8±1.40 | 74.04 |
| 3D InfoMax (Stärk et al., 2022) | 69.1±1.07 | 74.5±0.74 | 64.4±0.88 | 60.6±0.78 | 79.9±3.49 | 74.4±2.45 | 76.1±1.33 | 79.7±1.54 | 72.34 |
| GraphMVP (Liu et al., 2021) | 72.4±1.60 | 74.4±0.20 | 63.1±0.40 | 63.9±1.20 | 77.5±4.20 | 75.0±1.00 | 77.0±1.20 | 81.2±0.90 | 73.07 |
| MoleculeSDE (Liu et al., 2023) | 71.8±0.76 | 76.8±0.34 | 65.0±0.26 | 60.8±0.39 | 87.0±0.53 | **80.9±0.37** | 78.8±0.92 | 79.5±2.17 | 75.07 |
| MoleBLEND (Yu et al., 2023) | **73.0±0.81** | **77.8±0.89** | **66.1±0.03** | **64.9±0.35** | 87.6±0.75 | 77.2±2.38 | **79.0±0.89** | **83.7±1.46** | **76.16** |
| BIT | 74.3±0.81 | 78.1±0.91 | 66.4±0.18 | 64.8±0.47 | 91.3±1.21 | 79.4±0.87 | 80.2±0.67 | 84.5±0.85 | 77.38 |

Table 4: Ablation studies of MoDE and pre-training tasks.

| | **Pre-Training Tasks** | | **Backbone** | **Property** | | **Binding** | |
|---|---|---|---|---|---|---|---|
| | Token | Coordinate | MoDE | HIV ↑ | Tox21 ↑ | PDBbind (MAE) ↓ | DUD-E (AUC) ↑ |
| [1] | ✗ | ✗ | ✓ | 70.9 | 75.1 | 1.114 | 95.7 |
| [2] | ✓ | ✗ | ✓ | 78.5 | 76.1 | 1.016 | 97.1 |
| [3] | ✗ | ✓ | ✓ | 78.2 | 77.6 | 0.939 | 96.5 |
| [4] | ✓ | ✓ | ✗ | 78.3 | 76.9 | 0.977 | 98.0 |
| [5] | ✓ | ✓ | ✓ | **80.2** | **78.1** | **0.927** | **98.4** |

Following previous studies (Hu et al., 2020), we employ scaffold splitting to divide the dataset into training, validation, and test sets in an 8:1:1 ratio. We use the ROC-AUC as the evaluation metric and report the mean and standard deviation of the results obtained from 3 random seed runs. We compare BIT against the most representative molecular graph-based as well as multimodal pre-trained models. Detailed descriptions of the baselines are presented in Appendix B.3.

The performance of BIT, compared to competitive baselines, is summarized in Table 3. We observe that BIT outperforms the baselines on 6 out of 8 tasks, and achieves an overall relative improvement of 1.6% in terms of average ROC-AUC compared to the previous state-of-the-art result.

## 4.4 ABLATION STUDIES

**MoDE.** We conduct ablation experiments to investigate the impact of MoDE. As presented in Table 4, the integration of MoDE significantly boosts performance across various tasks, particularly in the binding affinity prediction task (PDBbind), where it is essential to encode both ligands and proteins concurrently while capturing the fine-grained inter-molecular interactions. Such enhancement is in line with our motivation to introduce MoDE.

**pre-training tasks.** We also perform ablation studies to analyze the contribution of different pre-training tasks, and the results are presented in Table 4. Eliminating either pre-training objective leads to pronounced declines in performance. We observe that coordinate denoising is indispensable for 3D representations, whereas masked token denoising is paramount for 2D representations. These results indicate that our unified pre-training is crucial and yields positive outcomes.

## 5 CONCLUSION AND FUTURE WORK

In this work, we take further strides towards general-purpose molecular modeling. We introduce BIT, a pre-trained foundation model, which is designed to encode molecules across various biochemical domains, including small molecules, proteins, and protein-ligand complexes, in different data formats, including 2D and 3D structures. Experimental results demonstrate that BIT excels across a broad spectrum of protein-ligand binding and molecular learning tasks. In our **future work**, we plan to work on fine-tuning BIT for structure-based molecular generation tasks, such as target protein binding (Luo et al., 2021) and molecular docking (Corso et al., 2022). We are also working on gathering more diverse real-world and synthetic protein-ligand complexes to facilitate the training of larger models.

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

## A    IMPLEMENTATION DETAILS OF TRANSFORMER-M

### A.1    PREDICTION HEAD FOR POSITION OUTPUT.

We use the SE(3) equivariant prediction head proposed in Shi et al. (2022):

$$\hat{\vec{\epsilon}}_i^k = (\sum_{v_j \in V} a_{ij} \Delta_{ij}^k \boldsymbol{X}_j^{(L)} \boldsymbol{W}_N^1) \boldsymbol{W}_N^2, \quad k = 0, 1, 2 \tag{6}$$

where $\boldsymbol{X}_j^{(L)}$ is the output of the last Transformer block, $a_{ij}$ is the attention score between atom $i$ and $j$ calculated by Eqn.3, $\Delta_{ij}^k$ is the k-th element of the directional vector $\frac{\vec{r}_i - \vec{r}_j}{\|\vec{r}_i - \vec{r}_j\|}$ between atom $i$ and $j$, and $\boldsymbol{W}_N^1 \in \mathbb{R}^{d \times d}$, $\boldsymbol{W}_N^2 \in \mathbb{R}^{d \times 1}$ are learnable weight matrices.

## B    EXPERIMENTAL DETAILS

### B.1    PDBBIND

**Evaluation Metrics.** Root Mean Square Error (RMSE), Mean Absolute Error (MAE) and Pearson correlation coefficient (R) are defined as:

$$RMSE = \sqrt{\frac{1}{|\mathcal{D}|} \sum_{i=1}^{|\mathcal{D}|} (\hat{y}_i - y_i)^2}, \; MAE = \frac{1}{|\mathcal{D}|} \sum_{i=1}^{|\mathcal{D}|} |\hat{y}_i - y_i| \tag{7}$$

$$R = \frac{\sum_{i=1}^{|\mathcal{D}|} (\hat{y}_i - \bar{\hat{y}})(y_i - \bar{y})}{\sqrt{\sum_{i=1}^{|\mathcal{D}|} (\hat{y}_i - \bar{\hat{y}})^2 (y_i - \bar{y})^2}} \tag{8}$$

$\hat{y}_i$ and $y_i$ respectively represent the predicted and experimental binding affinity of the $i$-th complex in dataset $\mathcal{D}$. The standard deviation (SD) is defined as follows:

$$SD = \sqrt{\frac{1}{|\mathcal{D}| - 1} \sum_{i=1}^{|\mathcal{D}|} [y_i - (a + b\hat{y}_i)]^2} \tag{9}$$

where $a$ and $b$ are the intercept and the slope of the regression line, respectively.

**Settings.** We fine-tune the pre-trained BIT on the PDBbind dataset. We use AdamW (Loshchilov & Hutter, 2018) as the optimizer and set its hyperparameter $\epsilon$ to 1e-8 and $(\beta_1, \beta_2)$ to (0.9,0.999). The gradient clip norm is set to 5.0. The peak learning rate is set to 1e-5. The total number of epochs is set to 120. The ratio of the warm-up steps to the total steps is set to 0.06. The batch size is set to 32. The dropout ratios for the input embeddings, attention matrices, and hidden representations are set to 0.0, 0.1, and 0.0 respectively. The weight decay is set to 0.0.

### B.2    DUD-E

**Settings.** We fine-tune the pre-trained BIT on the DUD-E dataset. We use AdamW (Loshchilov & Hutter, 2018) as the optimizer and set its hyperparameter $\epsilon$ to 1e-8 and $(\beta_1, \beta_2)$ to (0.9,0.999). The gradient clip norm is set to 5.0. The peak learning rate is set to 2e-4. The total number of epochs is set to 10. The ratio of the warm-up steps to the total steps is set to 0.06. The batch size is set to 16. The dropout ratios for the input embeddings, attention matrices, and hidden representations are set to 0.0, 0.1, and 0.0 respectively. The weight decay is set to 0.0.

### B.3    MOLECULENET

**Dataset.** The details of the 8 datasets used in this work are described below.

- BBBP: Blood-brain barrier penetration (BBBP) contains the ability of small molecules to penetrate the blood-brain barrier.

- Tox21: The dataset contains toxicity measurements of 8k molecules for 12 targets.
- ToxCast: This dataset is derived from toxicology data from in vitro high-throughput screening and contains toxicity measurements for 8k molecules against 617 targets.
- SIDER: The Side Effect Resource (SIDER) contains side effects of drugs on 27 system organs. These drugs are not only small molecules but also some peptides with molecular weights over 1000.
- ClinTox: This dataset contains the toxicity of the drug in clinical trials and the status of the drug for FDA approval.
- MUV: Maximum Unbiased Validation (MUV) is another subset of PubChem BioAssay, containing 90k molecules and 17 bioassays.
- HIV: This dataset contains 40k compounds with the ability to inhibit HIV replication.
- BACE: This dataset contains the results of small molecules as inhibitors of binding to human $\beta$-secretase 1 (BACE-1).

**Baselines.** We compare BIT against both molecular graph-based pre-trained models, including AttrMask (Hu et al., 2020), GraphCL (You et al., 2020), InfoGraph (Sun et al., 2020), GROVER (Rong et al., 2020), MolCLR (Wang et al., 2022a), GraphMAE (Hou et al., 2022) and Mole-BERT (Xia et al., 2023a), as well as multimodal pre-trained models, including 3D infoMax (Stärk et al., 2022), GraphMVP (Liu et al., 2021), MoleculeSDE (Liu et al., 2023), and MoleBLEND (Yu et al., 2023).

**Settings.** We use a grid search to find the best combination of hyperparameters for the molecular property prediction task. The specific search space is shown in Table 5. In all experiments, we choose the checkpoint with the lowest validation loss, and report the results on the test set run by that checkpoint.

Table 5: Search space for the MoleculeNet benchmark.

| Hyperparameter | Search space |
|---|---|
| Learning rate | [2e-5, 5e-5, 1e-4, 2e-4] |
| Batch size | [32, 64, 128, 256] |
| Warmup ratio | [0, 0.06] |

