# OpenReview forum: "General-purpose Pre-trained Model Towards Cross-domain Molecule Learning"
_ICLR.cc/2024/Conference — Submitted to ICLR 2024_

### Official Review · Reviewer_x6ZB · 2023-10-24

**Soundness:** 2 fair
**Presentation:** 2 fair
**Contribution:** 2 fair
**Rating:** 5
**Confidence:** 4

**Summary:**

The paper proposes a general protein pre-training model, i.e., Biomolecular Interaction Transformer (BIT) to process molecules and pocket-ligand complexes in a unified style. Specifically, the main block of BIT is based on Transformer-M, a previous pre-training model. To enhance the model’s ability of capturing multi- and inter-domain relationships, authors further incorporate Mixture-of-Domain-Experts (MoDE), i.e., separate feed-forward layers, for fusing molecule and pocket information better. Experiment results show that BIT achieves state-of-the-art performance in various downstream tasks, including binding affinity prediction, virtual screening, and molecular property prediction.

**Strengths:**

- BIT shows great performance in all downstream tasks listed in the paper.
- BIT can handle molecules and protein pockets in a unified way.
- BIT can work well with both 2D and 3D molecules.

**Weaknesses:**

- Contribution is minor. BIT combines the architecture of Transformer-M and Mixture-of-Domain-Experts technique, which both are from existing methods [1, 2, 3]. In addition, the way of combining the two is also a simple adaption. In summary, I appreciate that authors provide a strong method but I also believe that the contribution of this paper is not enough for acceptance.
- Experiments are not convincing enough:
1) Authors did not provide ablation studies to directly show the effectiveness of the main components, e.g., MoDE, of BIT.
2) The comparison between BIT and the main baseline, i.e., Transformer-M, may not be fair enough. As BIT adopts MoDE technique, it has more trainable parameters than the vanilla Transformer-M. Moreover, BIT uses not only small molecule data but also protein-ligand complex data in the pre-training stage, while Transformer-M only uses small molecule data.
3) The results of Transformer-M are not included in virtual screening and molecular property prediction benchmarks.
- Some important details are missing, e.g.,
1) In section 3.1, authors mentioned domain type embedding but without further description.
2) Also in section 3.1, authors introduce two special nodes, i.e., [M_VNode] and [P_VNodes]. It is unclear how to build up the input representation with these two nodes.
3) What is the specific value of noise scale controlling hyperparameter $\sigma$?

[1] Luo, S., Chen, T., Xu, Y., Zheng, S., Liu, T. Y., Wang, L., & He, D. (2022). One transformer can understand both 2d & 3d molecular data. arXiv preprint arXiv:2210.01765.

[2] Shazeer, N., Mirhoseini, A., Maziarz, K., Davis, A., Le, Q., Hinton, G., & Dean, J. (2017). Outrageously large neural networks: The sparsely-gated mixture-of-experts layer. arXiv preprint arXiv:1701.06538.

[3] Fan, Z., Sarkar, R., Jiang, Z., Chen, T., Zou, K., Cheng, Y., ... & Wang, Z. (2022). M³vit: Mixture-of-experts vision transformer for efficient multi-task learning with model-accelerator co-design. Advances in Neural Information Processing Systems, 35, 28441-28457.

**Questions:**

- In section 3.3, why only add noise to molecules when doing pre-training?
- In section 3.4.1, Why not use [P_VNode] or the combination of [M_VNode] and [P_VNode] as the representation of protein-ligand complexes?
- In table 3, why the PCBA dataset is not included in the table?

---

> ### Author Response · Authors · 2023-11-21
> **Response to Reviewer x6ZB (1/2)**
>
> We deeply appreciate the reviewer for the insightful and constructive comments!
>
> > Contribution is minor. BIT combines the architecture of Transformer-M and Mixture-of-Domain-Experts technique, which both are from existing methods [1, 2, 3]. In addition, the way of combining the two is also a simple adaption.
>
> We respectfully disagree with the reviewer that our contribution is minor. We acknowledge that the Transformer-M architecture and Mixture-of-experts technique have been verified in previous works, which we have extensively cited in section 3.2. We did not claim these or their combination as our contribution.
>
> We want to clarify that our main contribution is designing a general-purpose self-supervised framework for biomolecule modeling, as summarized in the Introduction. To the best of our knowledge, we are the first to propose a viable solution for this objective, and to learn fine-grained inter-molecular interactions with a simple backbone. Along the path to unified self-supervised pre-training, we make a series of key contributions:
>
> - A unified tokenization approach across diverse molecular domains, enabling the straightforward combination architecture of Transformer-M and MoDE. Existing works either train in one particular molecule domain, limiting the model robustness, or train separate specialized backbones for different data and token types, restricting model scalability.
> - Unified self-supervised pre-training objectives applicable to diverse pre-training data sources and downstream tasks. This enhances model versatility and performance across tasks.
> - A general methodology to unify the diverse structural and domain-specific information of biomolecules, enriching the training dataset to enhance model performance.
> - Reformulating virtual screening as a retrieval task, achieving state-of-the-art performance at ultra-fast inference speeds.
>
> In summary, we are the first to demonstrate the feasibility and potential of cross-domain molecular self-supervised learning with a single backbone. The efficiency and broad applicability of our framework allow it to outperform existing approaches. We believe this self-supervised pre-training framework is novel and valuable to the community.
>
> > 1. Authors did not provide ablation studies to directly show the effectiveness of the main components, e.g., MoDE, of BIT.
> > 2. The comparison between BIT and the main baseline, i.e., Transformer-M, may not be fair enough. As BIT adopts MoDE technique, it has more trainable parameters than the vanilla Transformer-M. Moreover, BIT uses not only small molecule data but also protein-ligand complex data in the pre-training stage, while Transformer-M only uses small molecule data.
> > 3. The results of Transformer-M are not included in virtual screening and molecular property prediction benchmarks.
>
> Thanks for your suggestion. We have conducted further experiments to investigate the impact of MoDE following the suggestion. We compare three backbones, all trained on the same samples including both small molecules and protein-ligand complexes. One backbone is the vanilla Transformer-M. The second is Transformer-M with a larger FFN (2 $\times$ hidden size), having the same parameters as BIT. The third is our proposed BIT. Note that we selected two representative datasets (i.e., a small Tox21 dataset and a large HIV dataset) from MoleculeNet for the ablation study on molecular property prediction, due to time constraints. The results are presented in the table below. We found that introducing MoDE significantly improves the performance, especially on PDBbind where both small molecules and proteins could be encoded simultaneously. This verifies the benefit of introducing MoDE.
>
> As for the results of Transformer-M not being included in virtual screening and molecular property prediction benchmarks, this is because Transformer-M did not provide the specific experimental results mentioned above, so we were unable to make a direct fair comparison with it.
>
> | Backbone | HIV (AUC) $\uparrow$ | Tox21 (AUC) $\uparrow$ | PDBbind (MAE) $\downarrow$ | DUD-E (AUC) $\uparrow$ |
> |---------|---------|---------|--------|--------|
> | Transformer-M | 78.3 | 76.9 | 0.977 | 98.0 |
> | Transformer-M (2 $\times$ hidden size) | 77.8  | 77.5 | 0.952 | 97.8 |
> | BIT | **80.2** | **78.1** | **0.927** | **98.4** |

---

> ### Author Response · Authors · 2023-11-21
> **Response to Reviewer x6ZB (2/2)**
>
> Thank you for pointing out our unclear description. We will revise the writing to provide a more comprehensive description.
>
> > In section 3.1, authors mentioned domain type embedding but without further description.
>
> We apologize for our unclear description. The domain type embedding, i.e. the molecule domain embedding and protein domain embedding $M_{type}, P_{type} \in R^d$, are two learnable vectors used to indicate the domain type of each atom token. They have the same length as atom feature embedding and structural positional embedding. These three embeddings are summed up as input to the self-attention module.
>
> > Also in section 3.1, authors introduce two special nodes, i.e., [M_VNode] and [P_VNode]. It is unclear how to build up the input representation with these two nodes.
>
> We apologize for our unclear description. These two special node representations function similarly to the [CLS] token in BERT. However, unlike BERT, each domain expert in our model has its own distinct indicator token. [M_VNode] is inserted at the beginning of the molecule representation sequence, while [P_VNode] is inserted at the beginning of the protein representation sequence. We make the connection between [M_VNode] and each molecule atom individually, and also make the connection between [P_VNode] and each protein atom individually. Both special node embeddings are randomly initialized and updated during training like regular node representation.
>
> > What is the specific value of noise scale controlling hyperparameter $\sigma$
>
> $\sigma$ is set to 0.2, the same value used in Transformer-M.
>
> > In section 3.3, why only add noise to molecules when doing pre-training?
>
> Since proteins tend to be relatively rigid in most cases, generative models typically assume knowledge of the bound protein structure [1]. This is a widely accepted assumption in current molecular docking research [2]. Therefore, we adopt this approach in our work and maintain a fixed protein pocket during ligand recovery, as done in related studies [2,3].
>
> [1] Pagadala, Nataraj S., Khajamohiddin Syed, and Jack Tuszynski. "Software for molecular docking: a review." Biophysical reviews 9 (2017): 91-102.
>
> [2] Corso, Gabriele, et al. "DiffDock: Diffusion Steps, Twists, and Turns for Molecular Docking." The Eleventh International Conference on Learning Representations. 2023.
>
> [3] Guan, Jiaqi, et al. "3D Equivariant Diffusion for Target-Aware Molecule Generation and Affinity Prediction." The Eleventh International Conference on Learning Representations. 2023.
>
> > In section 3.4.1, Why not use [P_VNode] or the combination of [M_VNode] and [P_VNode] as the representation of protein-ligand complexes?
>
> In the representation of the protein-ligand complex, [M_VNode] is positioned as the initial token, whereas the placement of [P_VNode] is more flexible. Therefore, using [M_VNode] simplifies engineering implementation. Additionally, early empirical studies demonstrated that using [M_VNode] either individually or in combination with [P_VNode] resulted in similar learning performance.
>
> > In table 3, why the PCBA dataset is not included in the table?
>
> For molecular property prediction, we compare against the most representative 2D and 3D pretraining baselines, and report their official results from [4]. We exclude PCBA since most baselines have not reported results on this dataset. However, BIT still achieves the state-of-the-art performance among available records, as shown below:
>
> | Backbone | PCBA (AUC) $\uparrow$ |
> |---------|---------|
> | PretrainGNN | 86.0 |
> | GROVER | 83.0 |
> | BIT | **88.3** |
>
> [4] Yu, Qiying, et al. "Unified Molecular Modeling via Modality Blending." arXiv preprint arXiv:2307.06235v1 (2023).
>
> **We hope our response can alleviate your concerns. Please let us know if you have any additional questions.**

---

> ### Comment · Reviewer_x6ZB · 2023-11-22
>
> I appreciate your detailed responses. The additional experiments have alleviated my concerns that the model's performance is solely due to increased parameters and data. Furthermore, I concur that developing a general-purpose model for both protein pockets and small molecules is an innovative concept. However, I still maintain that BIT lacks technical novelty due to the following reasons:
>
> - **Unified Tokenization Approach**: The tokenization method used in BIT is simply a standard approach that involves treating molecules at the atom level. This approach has been employed in previous studies, such as Uni-Mol and Transformer-M. Therefore, I do not consider it to be a novel technique. Additionally, I was unable to locate a detailed explanation of the tokenization approach in the paper.
>
> - **Unified Self-Supervised Pre-training Objectives**: Similar to the tokenization approach, the pre-training objectives of BIT are aligned to Transformer-M. It is hard to say the objectives are novel.
>
> - **General Methodology to Unify Domain-specific Information**: Because MoDE is very similar to MOE, the general methodology should be credited to MOE, which was proposed earlier.
>
> - **Reformulating Virtual Screening as a Retrieval Task**: DrugCLIP [1] has previously reformulated virtual screening as a retrieval task, and has provided a more comprehensive evaluation of this approach.
>
> In conclusion, I believe that BIT is a well-implemented model, but it lacks a deep understanding. I therefore maintain my original score.
>
> [1] Gao B, Qiang B, Tan H, et al. DrugCLIP: Contrastive Protein-Molecule Representation Learning for Virtual Screening. NeurIPS 2023

---

> > ### Author Response · Authors · 2023-11-22
> > **Response to Reviewer x6ZB**
> >
> > We deeply appreciate the reviewer for the detailed feedback.
> >
> > > - Unified Tokenization Approach.
> > >
> > > - General Methodology to Unify Domain-specific Information.
> >
> > We acknowledge that the tokenization method has been employed in the studies you mentioned, but our motivation is completely different from theirs. Based on this tokenization method, we hope to design a unified model that can share general atom-level knowledge and capture domain-level specificity to handle molecules from diverse domains. To be precise, Transformer-M did not consider a specialized design to capture domain-level specificity. Directly applying such a model leads to limited transferability from one domain to another. One trivial solution is to separately train two models on small molecule data and protein pocket data, as down in Uni-Mol. However, this method can only capture the intra-molecular interactions and fails to capture the fine-grained inter-molecular interactions. Similarly, DrugCLIP, which adopts a dual-encoder architecture (i.e., two Uni-Mol backbones) to encode pockets and ligands separately, still maintains this limitation. Therefore, DrugCLIP is primarily used for virtual screening, restricting its application to more binding-related tasks that require inter-molecular interactions, such as binding affinity prediction. In contrast, our proposed model can address these issues and be applied to a wider range of downstream tasks.
> >
> > To further achieve our goal, we introduced MoDE. We do not simply adopt MOE, but the design choice is determined based on the following general methodology: similar to the two structural channels (i.e., bias terms in the self-attention module) in Transformer-M, BIT can be interpreted as sharing the flexible self-attention module for unified modeling, while utilizing separate parameters (i.e., MoDE) to capture the specificity among different inputs. We argue that structural information is relative between atoms, making it suitable to be captured within the self-attention module, while domain information is at the instance level, thus making it more appropriate to be captured within the Feed-Forward Networks. In light of the above consideration, MoDE can be harmoniously integrated with Transformer-M, expanding the application scope and significantly enhancing performance.
> >
> > > Unified Self-Supervised Pre-training Objectives.
> >
> > Our pre-training procedure is not aligned with Transformer-M. Our insight is to propose a unified pre-training objective applicable to various domain data (e.g., unbound molecules and protein-ligand complexes), which involves restoring the molecules (ligands) with low-energy structures from the corrupted states, depending on the biomolecular context. The same pre-training objective makes the training process scaling-up friendly. Experiments have demonstrated that our pre-training objective aligns small molecules and proteins effectively.
> >
> > Moreover, compared to Transformer-M, our approach offers greater generalizability and applicability for the following two reasons: 1）supervised pre-training based on the homo-lumo gap prediction may lead to negative transfer, and 2) there is extremely limited data with both 3D co-crystal structures and experimentally determined binding affinity values (approximately 23,000 in PDBbind).
> >
> > > Reformulating Virtual Screening as a Retrieval Task.
> >
> > Indeed, DrugCLIP's earliest public version appeared on Arxiv on 10 Oct, later than our submission deadline (29 Sep), so we did not follow them. We introduce additional experiments to compare BIT and DrugCLIP. In addition to the DUD-E dataset, we test the BIT model, which is fine-tuned on DUD-E, on the unbiased external test set LIT-PCBA. Note that we exclude all the targets present in LIT-PCBA from the training set to prevent data leakage. Our method consistently outperforms DrugCLIP. We argue that our proposed unified framework indeed helps the model learn more knowledge on molecular interactions. This observation also aligns well in the multimodal domain (e.g., vision-language pre-training) [1,2], where pre-training objectives that require deep fusion of image and text can lead to better performance on image-text retrieval tasks than CLIP.
> >
> > [1] Li, Junnan, et al. "Align before fuse: Vision and language representation learning with momentum distillation." NeurIPS, 2021.
> >
> > [2] Li, Junnan, et al. "Blip: Bootstrapping language-image pre-training for unified vision-language understanding and generation." ICML, 2022.
> >
> > DUD-E:
> > |  | AUROC| $RE_{0.005}$| $RE_{0.01}$| $RE_{0.02}$| $RE_{0.05}$|
> > |-|-|-|-|-|-|
> > | DrugCLIP | 96.6 | 118.10 | 67.17 | 37.17 | 16.59 |
> > | BIT |**98.4** |**141.63** | **78.31** | **42.43** | **18.44** |
> >
> > LIT-PCBA:
> > |  | AUROC| $EF_{0.005}$| $EF_{0.01}$| $EF_{0.05}$|
> > |-|-|-|-|-|
> > | DrugCLIP | 57.17 | 8.56 | 5.51 | 2.27 |
> > | BIT | **61.44** | **10.07** | **5.80** | **2.87** |
> >
> > **We hope our responses have addressed your concerns. Please let us know if you have any additional questions.**

---

### Official Review · Reviewer_uSoZ · 2023-10-30

**Soundness:** 4 excellent
**Presentation:** 4 excellent
**Contribution:** 3 good
**Rating:** 8
**Confidence:** 4

**Summary:**

The “General purpose pre-trained model…” paper proposes a Biomolecular Interaction Transformer BIT, which is to have a multi-modal training on molecules, together with protein—ligand matching, with 2-D and 3-D structures. The model includes a pre-training.

In my opinion, this is a valuable paper, presenting a well-defined model, together with well-designed experiments. The BIT model might not be revolutionary, but is a piece of a very solid work, and well performing. I opt for accepting this paper for the conference.

**Strengths:**

1. An advanced model, encompassing 2-D and 3-D structures for molecules, proteins, and ligand—protein interaction modelling, with multi-modal training. The model can be tuned.
2. Well-defined multi-modal representation learning employing a Transformer model (a Transformer-M model of Luo et al.) with independent tuning of proposed BIT for different knowledge domains.
3. A very good graphical abstract is given on page 2, showing in detail the proposed architecture. Clear presentation. All this increase the paper readability greatly.

**Weaknesses:**

1. Some generalization of the model to other areas would be welcome.

**Questions:**

1. In the comparison tables, the models (usually proposed BIT) with the best mean have values given in boldface. Are all the models trained on the same data-sets? For some predicted features, the differences between BIT and some other models are large, even though some of them are Transformers too. Are the optimal values for BIT the result of the proposed BIT architecture, the fine-tuning on different modes (the multi-modality), different data sets, better pre-training, or something other? The discussion on the comparison to other approaches is needed in the conclusions/discussion section.
2. Please correct the spelling of some words. Just as well, please correct the editing of mathematical expressions. E.g. in equations (1) and (2) the equal signs = should be aligned ;-)
3. Your paper and model is strictly molecule learning oriented. Do you think that the general approach can be used in other sciences? E.g. in biological experiments on cancerous cells and the impact of some sort of certain treatments, which would imply modifications in the course of the operation?

---

> ### Author Response · Authors · 2023-11-21
> **Response to Reviewer uSoZ**
>
> We deeply appreciate the reviewer for the insightful and constructive comments!
>
> > In the comparison tables, the models (usually proposed BIT) with the best mean have values given in boldface. Are all the models trained on the same data-sets? For some predicted features, the differences between BIT and some other models are large, even though some of them are Transformers too. Are the optimal values for BIT the result of the proposed BIT architecture, the fine-tuning on different modes (the multi-modality), different data sets, better pre-training, or something other? The discussion on the comparison to other approaches is needed in the conclusions/discussion section.
>
> Thanks for pointing this out. You are right that our method and other pre-training baselines are inconsistent in terms of the pre-training data and pre-training objectives. Being able to leverage more cross-domain data is an advantage of our proposed BIT. For downstream tasks, we report the official results of baselines and follow the same experimental settings as baselines to ensure a fair comparison. We will discuss this issue in the revised paper.
>
> You provided us with an excellent suggestion to better isolate the benefit of our backbone modeling approach through controlled experiments. Thus, we introduce ablation experiments to investigate the factors influencing performance. Using the same pre-training data, we remove select pre-training objectives or train different backbones with identical procedures to quantify their impact. Note that we selected two representative datasets (i.e., a small Tox21 dataset and a large HIV dataset) from MoleculeNet for the ablation study on molecular property prediction, due to time constraints. As summarized in the following tables, we find 1) our two pre-training objectives are necessary and effective, and 2) introducing MoDE significantly improves performance. In the revision, we will incorporate these ablation studies to provide a controlled comparison.
>
> | Pre-training objectives | HIV (AUC) $\uparrow$ | Tox21 (AUC) $\uparrow$ | PDBbind (MAE) $\downarrow$ | DUD-E (AUC) $\uparrow$ |
> |---------|---------|--------|--------| --------|
> | BIT | **80.2** | **78.1** | **0.927** | **98.4** |
> | w/o Masked token denoising | 78.2 | 77.6 | 0.939 | 96.5 |
> | w/o Coordinate denoising | 78.5 | 76.1 | 1.016 | 97.1 |
> | w/o pre-training | 70.9 | 75.1 | 1.114 | 95.7 |
>
> | Backbone | HIV (AUC) $\uparrow$ | Tox21 (AUC) $\uparrow$ | PDBbind (MAE) $\downarrow$ | DUD-E (AUC) $\uparrow$ |
> |---------|---------|---------|--------|--------|
> | BIT | **80.2** | **78.1** | **0.927** | **98.4** |
> | w/o MoDE | 78.3 | 76.9 | 0.977 | 98.0 |
> | w/o MoDE (2 $\times$ hidden size) | 77.8  | 77.5 | 0.952 | 97.8 |
>
> > Please correct the spelling of some words. Just as well, please correct the editing of mathematical expressions. E.g. in equations (1) and (2) the equal signs = should be aligned ;-)
>
> Thanks for your patience and helpful reminder. We will carefully revise our writing.
>
> > - Some generalization of the model to other areas would be welcome.
> >
> > - Your paper and model is strictly molecule learning oriented. Do you think that the general approach can be used in other sciences? E.g. in biological experiments on cancerous cells and the impact of some sort of certain treatments, which would imply modifications in the course of the operation?
>
> Thank you for acknowledging and supporting our work. Our current model focuses on general-purpose biomolecule learning. It operates at the molecular resolution level and could potentially be adopted for problems determined by physical structure. However, progress remains to be made in order to handle more general biological problems that cannot be fully explained through molecule interactions.
>
> The idea of multi-domain modeling is powerful and could be applied to other scientific domains to capture correlations between different data domains, such as precision therapy (cell lines and molecules) and text-guided protein design (text and proteins).
>
> We plan to pursue more general AI capabilities step by step. The first step is building a foundation model for biomolecules and biochemical reactions. After this, we intend to integrate additional multi-domain data including images and text, which could enable solving more general biomedical problems. This feedback has been invaluable for identifying opportunities and challenges moving forward.

---

> > ### Comment · Reviewer_uSoZ · 2023-12-02
> > **Assessment of the presentation authors' 942 responses to my questions generally satisfy me.**
> >
> > I am satisfied with the submission 942 answers to my questions, questions of which there were not many. Particularly, I am happy with your addition of the ablations sections, that I can see most of the reviewers felt a need too. I am only minimally disappointed that you have not been able to indicate, let alone add to the conclusions or comments, more precise indications of where the BIT approach could be used.
> > Still, the whole model does not contribute much, and my first assessments were probably overly optimistic. Thus, I cannot increase my ratings.

---

### Official Review · Reviewer_dgYT · 2023-11-01

**Soundness:** 2 fair
**Presentation:** 3 good
**Contribution:** 2 fair
**Rating:** 5
**Confidence:** 4

**Summary:**

In this study, a new pre-trained transformer, BIT, is introduced for processing small molecules, proteins, and ligand-protein complexes. The architecture is based on Transformer-M, which is a transformer architecture that enables the processing of 2D and 3D structures. The main architectural innovation introduced in this work is the use of Mixture-of-Domain-Experts (MoDE) that replaces feed-forward layers, allowing for different processing of small molecules and macromolecules. Two pre-training methods, masking and coordinate denoising, are used to improve the performance of this model. BIT can be utilized for various molecular tasks, including molecular property prediction, structure-based virtual screening, and binding affinity prediction. The experimental section shows that BIT outperforms similar approaches in all three tasks.

**Strengths:**

- The proposition to use Mixture-of-Domain-Experts in order to process both small molecules and proteins and protein-ligand complexes is interesting. This way, the transformer can be (pre)trained using more data with high diversity.
- Two pre-training methods are implemented, and the strong performance of the pre-trained transformer is demonstrated in the experiments.
- The motivation of the paper is clear, and the methodology and experiments are easy to follow.
- This work has some significant applications in the molecular modeling domain, especially in structure-based drug design. Because BIT can process both small molecules and proteins, the application domain is very broad. The significance of the study is corroborated by the strong performance in the molecular property prediction, binding affinity prediction, and structure-based virtual screening tasks.

**Weaknesses:**

- The main novelty of the paper is the introduction of MoDE in order to process data from diverse molecular domains. In my opinion, the paper lacks a proper evaluation of what these experts learn. For example, do molecule and pocket experts learn similar weights, or are there significant differences? What is the performance of this model when only one type of feed-forward layer is used (like in Transformer-M), but the same pretraining procedure is applied?
- The Authors propose to use two pre-training objectives. It would be interesting to see what is the impact of each of them. Why did the Authors decide to use a different pre-training procedure than used in Transformer-M? The experimental tables are missing the results achieved for the non-pretrained model.
- The choice of the models used in the experiments seems arbitrary. For example, why are different models used for binding affinity prediction and molecular property prediction? GROVER could be used in both scenarios. Why are some of these models not pre-trained, while the original works provide pre-training procedures (and sometimes also the pre-trained weights), e.g. GROVER and MAT?
- Finally, another benchmark for structure-based virtual screening would be helpful in assessing the performance of the proposed method. It has been shown, that decoys contain hidden biases that can impact the performance of deep learning methods [1].

[1] Chen, Lieyang, et al. "Hidden bias in the DUD-E dataset leads to misleading performance of deep learning in structure-based virtual screening." PloS one 14.8 (2019): e0220113.

In conclusion, the paper introduces some new ideas (mixing of domains and a different pre-training procedure), but the presented results do not support these design choices. It is not clear which novelties contribute most to the strong performance of BIT.

Minor comments (not impacting my evaluation):
- The first paragraph of the Introduction section: "depend on these information"

**Questions:**

1. For binding affinity prediction, did you consider measuring non-linear correlation, e.g. using the Spearman correlation coefficient? This evaluation metric could be better at showing which methods can correctly prioritize compounds with strong affinity.
2. Do you plan to publish the code for better reproducibility of your method?

---

> ### Author Response · Authors · 2023-11-21
> **Response to Reviewer dgYT (1/2)**
>
> We deeply appreciate the reviewer for the insightful and constructive comments!
>
> > The main novelty of the paper is the introduction of MoDE in order to process data from diverse molecular domains. In my opinion, the paper lacks a proper evaluation of what these experts learn. For example, do molecule and pocket experts learn similar weights, or are there significant differences? What is the performance of this model when only one type of feed-forward layer is used (like in Transformer-M), but the same pretraining procedure is applied?
>
> Thanks for pointing this out. To further investigate the impact of MoDE, we introduce additional experiments. In these experiments, we use the same pre-training procedure to train two additional backbones. One backbone omits MoDE and uses a shared FFN, while the other also omits MoDE but utilizes a larger FFN (2 $\times$ hidden size) to match BIT in parameter scale. Note that we selected two representative datasets (i.e., a small Tox21 dataset and a large HIV dataset) from MoleculeNet for the ablation study on molecular property prediction, due to time constraints. The results are presented in the table below. We find that introducing MoDE significantly improves performance, especially on PDBbind where both small molecules and proteins could be encoded simultaneously. This aligns with our motivation for introducing MoDE. Furthermore, the performance gains cannot be attributed solely to increased parameters, as using a larger FFN alone did not consistently improve performance across tasks.
>
> | Backbone | HIV (AUC) $\uparrow$ | Tox21 (AUC) $\uparrow$ | PDBbind (MAE) $\downarrow$ | DUD-E (AUC) $\uparrow$ |
> |---------|---------|---------|--------|--------|
> | BIT | **80.2** | **78.1** | **0.927** | **98.4** |
> | w/o MoDE | 78.3 | 76.9 | 0.977 | 98.0 |
> | w/o MoDE (2 $\times$ hidden size) | 77.8  | 77.5 | 0.952 | 97.8 |
>
> > The Authors propose to use two pre-training objectives. It would be interesting to see what is the impact of each of them. Why did the Authors decide to use a different pre-training procedure than used in Transformer-M? The experimental tables are missing the results achieved for the non-pretrained model.
>
> Thanks for your suggestion. To examine the impact of the two pre-training objectives, we introduce additional experiments by eliminating select objectives to study their effectiveness. As shown in the table below, the pretrain-then-finetune paradigm is significantly superior to direct supervised training for a Transformer model. We also observe a notable performance decrease when removing either pre-training objective, particularly coordinate denoising for PDBbind (2D+3D) and masked token denoising for molecular property prediction (2D). These results indicate that both pre-training objectives are crucial and yield positive outcomes.
>
> We did not use the pre-training procedure from Transformer-M for two reasons: 1) supervised pre-training based on the homo-lumo gap prediction may lead to negative transfer, and 2) there is extremely limited data with both 3D co-crystal structures and experimentally determined binding affinity values (approximately 23,000 in PDBbind). Our aim is to build a more versatile model using more unsupervised data, so we adopt a purely self-supervised pre-training approach.
>
> | Pre-training objectives | HIV (AUC) $\uparrow$ | Tox21 (AUC) $\uparrow$ | PDBbind (MAE) $\downarrow$ | DUD-E (AUC) $\uparrow$ |
> |---------|---------|--------|--------| --------|
> | BIT | **80.2** | **78.1** | **0.927** | **98.4** |
> | w/o Masked token denoising | 78.2 | 77.6 | 0.939 | 96.5 |
> | w/o Coordinate denoising | 78.5 | 76.1 | 1.016 | 97.1 |
> | w/o pre-training | 70.9 | 75.1 | 1.114 | 95.7 |

---

> > ### Author Response · Authors · 2023-11-21
> > **Response to Reviewer dgYT (2/2)**
> >
> > > The choice of the models used in the experiments seems arbitrary. For example, why are different models used for binding affinity prediction and molecular property prediction? GROVER could be used in both scenarios. Why are some of these models not pre-trained, while the original works provide pre-training procedures (and sometimes also the pre-trained weights), e.g. GROVER and MAT?
> >
> > We apologize for the unclear description of the baseline. The choice of baselines follows these rules: for molecular property prediction, we choose the most representative 2D and 3D pre-train baselines and report their official results from [1]; for binding affinity prediction, we report the official results of state-of-the-art GNN-based baselines from [2], and also include the recently published pre-trained methods Transformer-M and MBP [3]. We will provide a clear description in the corresponding tables. Neither GROVER nor MAT provided the specific experimental results mentioned above, so we were unable to make a direct fair comparison with them.
> >
> > [1] Yu, Qiying, et al. "Unified Molecular Modeling via Modality Blending." arXiv preprint arXiv:2307.06235v1 (2023).
> >
> > [2] Li, Shuangli, et al. "Structure-aware interactive graph neural networks for the prediction of protein-ligand binding affinity." SIGKDD. 2021.
> >
> > [3] Yan, Jiaxian, et al. "Multi-task Bioassay Pre-training for Protein-ligand Binding Affinity Prediction." arXiv preprint arXiv:2306.04886 (2023).
> >
> > > Finally, another benchmark for structure-based virtual screening would be helpful in assessing the performance of the proposed method. It has been shown, that decoys contain hidden biases that can impact the performance of deep learning methods.
> >
> > Thank you for the recommendation. To further validate BIT’s performance, we follow the suggestion in the comment and test the BIT model, which is fine-tuned on DUD-E, on the unbiased external test set LIT-PCBA [4]. Note that we exclude all the targets present in LIT-PCBA from the training set to prevent data leakage. BIT still outperforms the conventional docking method Glide-SP, the supervised deep learning method Planet, and the pre-trained method DrugCLIP, as illustrated below:
> >
> > |  | AUROC $\uparrow$ |  $EF_{0.005}$ $\uparrow$ | $EF_{0.01}$ $\uparrow$ | $EF_{0.05}$ $\uparrow$ |
> > |---------|---------|--------|--------| --------|
> > | Glide-SP | 53.15 | 3.17 | 3.41 | 2.01 |
> > | Planet   | 57.31 | 4.64 | 3.87 | 2.43 |
> > | DrugCLIP | 57.17 | 8.56 | 5.51 | 2.27 |
> > | BIT      | **61.44** | **10.07** | **5.80** | **2.87** |
> >
> > [4] Tran-Nguyen, Viet-Khoa, Célien Jacquemard, and Didier Rognan. "LIT-PCBA: an unbiased data set for machine learning and virtual screening." Journal of chemical information and modeling 60.9 (2020): 4263-4273.
> >
> > > For binding affinity prediction, did you consider measuring non-linear correlation, e.g. using the Spearman correlation coefficient? This evaluation metric could be better at showing which methods can correctly prioritize compounds with strong affinity.
> >
> > We have considered using non-linear correlation to evaluate the ‘scoring power' on PDBbind. However, as established in the standard comparative assessment of scoring functions (CASF) benchmark [5], such metrics are better suited to quantifying 'ranking power’, where multiple ligands are ranked against the same target protein. Since PDBbind contains only one ligand per unique target protein, the non-linear correlation is not applicable in this case. Thus, we quantify 'scoring power' by reporting key regression metrics - Root Mean Square Error (RMSE), Mean Absolute Error (MAE), Pearson's correlation coefficient (R), and the standard deviation (SD), following precedents from previous work [6], to ensure a fair comparison.
> >
> > [5] Su, Minyi, et al. "Comparative assessment of scoring functions: the CASF-2016 update." Journal of chemical information and modeling 59.2 (2018): 895-913.
> >
> > [6] Stepniewska-Dziubinska, Marta M., Piotr Zielenkiewicz, and Pawel Siedlecki. "Development and evaluation of a deep learning model for protein–ligand binding affinity prediction." Bioinformatics 34.21 (2018): 3666-3674.
> >
> > > Do you plan to publish the code for better reproducibility of your method?
> >
> > Yes. The code will be made publicly available upon acceptance of the paper.
> >
> > **We hope our response can alleviate your concerns. Please let us know if you have any additional questions.**

---

### Official Review · Reviewer_2Mm5 · 2023-11-04

**Soundness:** 2 fair
**Presentation:** 3 good
**Contribution:** 2 fair
**Rating:** 3
**Confidence:** 4

**Summary:**

This paper presents a multitask Transformer architecture for molecular interaction learning called Biomolecular Interaction Transformer (BIT)  and a self-supervised learning objective of coordinate denoising and masked token denoising. This approach allows the model to learn representations of biomolecules of both 2D and 3D structures. Moreover, the paper introduces the Mixture-of-Domain-Experts (MoDE) module, which enables the model to learn from biomolecules belonging to different chemical domains (proteins and ligands).

**Strengths:**

+ The paper is well motivated and the writing is clear.
+ The proposed approach focuses on protein-ligand interactions, which is of paramount importance for drug discovery and can potentially lead to more efficient and effective therapeutic solutions.

**Weaknesses:**

- The pretraining process appears to be confined to proteins and small molecules. This limited scope raises questions about the model's applicability to a broader range of biomolecules and interactions.
- The proposed Transformer backbone is invariant to geometric transformations, which may limit its expressive power compared to SE(3) or E(3) equivariant architectures.
- The model only considers the binding pocket segment of proteins—while computationally efficient, may not be practically feasible without costly simulations to identify these segments. This reliance on prior knowledge or expensive computations could limit the accessibility and scalability in practical applications.
- The strategy of pre-training the model on both equilibrium structures of molecules and higher-energy protein-ligand complexes may be conceptually problematic. These two types of data represent vastly different energy states, and it is unclear what meaningful semantic learning can be achieved by pretraining them together. This may lead to a model that does not adequately distinguish between the distinct energetic landscapes of the two systems.  Also, the scales of the two training sources are not balanced, where PCQM4Mv2 is significantly larger than Q-BioLiP, but the model performance on molecular property prediction is not very significant compared to the baselines, which warrants a further examination.
- There is a lack of a detailed explanation for how positional encodings of 2D and 3D structures are integrated into the model, which leaves a gap in understanding the full architecture and mechanics of the model.

**Questions:**

Please see the above weaknesses section.

---

> ### Author Response · Authors · 2023-11-21
> **Response to Reviewer 2Mm5 (1/2)**
>
> We deeply appreciate the reviewer for the insightful and constructive comments!
>
> > The pretraining process appears to be confined to proteins and small molecules. This limited scope raises questions about the model's applicability to a broader range of biomolecules and interactions.
>
> Thanks for pointing this out. A key advantage of our proposed method over existing works is indeed its broader applicability. Existing methods typically use different tokens for different biomolecules, such as amino acids for proteins and atoms for small molecules. This requires designing specialized encoders for each type of biomolecule, limiting the applicability. Our method instead employs the same token (i.e., atom) for all biomolecule types and uses Mixture-of-Domain-Experts (MoDE) to capture domain specificity. By simply adding corresponding domain expert FFNs, our approach can be readily applied to any biomolecule and interaction.
>
> The reason we focus on proteins and small molecules in this paper is that protein-molecule binding-related tasks are typical applications with sufficient structural data. This aligns with the main focus of most related works in structure-based drug design and enables a direct comparison to the previous methods in the field. However, as the reviewer points out, investigating more diverse, high-quality biomolecules for pre-training would be valuable future work to further boost the performance and applicability of our model.
>
> > The proposed Transformer backbone is invariant to geometric transformations, which may limit its expressive power compared to SE(3) or E(3) equivariant architectures.
>
> Thanks for your thoughtful reminder. We have exactly the same question ourselves when designing the backbone structures. Ultimately, we choose our Transformer backbone over the equivariant architectures for the following reasons: While equivariant architectures efficiently encode rotational and translational symmetries for molecules, their computational overhead becomes prohibitive for large-scale pre-training. For our goal of pre-training a sizable model using atom tokens at scale, training scalability and stability are more crucial. Furthermore, sufficient training data equipped with augmentation techniques can compensate for the lack of built-in equivariance for the Transformer. Empirical results across tasks confirm our backbone's superior performance as expected. Though promising, equivariant networks' current challenges make our backbone a better fit for now.
>
> > The model only considers the binding pocket segment of proteins—while computationally efficient, may not be practically feasible without costly simulations to identify these segments. This reliance on prior knowledge or expensive computations could limit the accessibility and scalability in practical applications.
>
> This is true. Pocket searching falls outside the scope of this paper, which is focused on training a cross-domain pre-trained model for biomolecules. For the binding-related task, we follow the standard protocol from mainstream approaches [1,2] and the competing methods [3], which assume the binding pocket as given. Pocket searching remains an active research area that may require distinct methodological considerations. Machine learning also has demonstrated potential in this domain, as evidenced by recent approaches [4,5].
>
> [1] Trott, Oleg, and Arthur J. Olson. "AutoDock Vina: improving the speed and accuracy of docking with a new scoring function, efficient optimization, and multithreading." Journal of computational chemistry 31.2 (2010): 455-461.
>
> [2] Koes, David Ryan, Matthew P. Baumgartner, and Carlos J. Camacho. "Lessons learned in empirical scoring with smina from the CSAR 2011 benchmarking exercise." Journal of chemical information and modeling 53.8 (2013): 1893-1904.
>
> [3] Li, Shuangli, et al. "Structure-aware interactive graph neural networks for the prediction of protein-ligand binding affinity." Proceedings of the 27th ACM SIGKDD Conference on Knowledge Discovery & Data Mining. 2021.
>
> [4] Krivák, Radoslav, and David Hoksza. "P2Rank: machine learning based tool for rapid and accurate prediction of ligand binding sites from protein structure." Journal of cheminformatics 10 (2018): 1-12.
>
> [5] Gainza, Pablo, et al. "Deciphering interaction fingerprints from protein molecular surfaces using geometric deep learning." Nature Methods 17.2 (2020): 184-192.

---

> > ### Author Response · Authors · 2023-11-21
> > **Response to Reviewer 2Mm5 (2/2)**
> >
> > > The strategy of pre-training the model on both equilibrium structures of molecules and higher-energy protein-ligand complexes may be conceptually problematic. These two types of data represent vastly different energy states, and it is unclear what meaningful semantic learning can be achieved by pretraining them together. This may lead to a model that does not adequately distinguish between the distinct energetic landscapes of the two systems. Also, the scales of the two training sources are not balanced, where PCQM4Mv2 is significantly larger than Q-BioLiP, but the model performance on molecular property prediction is not very significant compared to the baselines, which warrants a further examination.
> >
> > We understand the reviewer’s concern is that the equilibrium structures of molecules and protein-ligand complexes may inhabit different energy states, as the same molecule may have different conformations in different biomolecule contexts, which may confuse the model. However, this is not necessarily problematic. Even though these are two distinct energy landscapes, our pre-training objective remains consistent across different types of pre-training data in restoring the low-energy structures from the corrupted ones depending on the biomolecule context. The issue is analogous to polysemy in NLP where a word can have multiple semantic meanings depending on the context. With sufficient training data, the model can learn context-aware representations automatically. Currently, without NLP-scale data, our solution is using separate domain experts for different biomolecules. We have also experimented with a unified Transformer architecture without MoDE, but the results were unsatisfactory as demonstrated below - the reviewer's observation may explain why. Note that we selected two representative datasets (i.e., a small Tox21 dataset and a large HIV dataset) from MoleculeNet for the ablation study on molecular property prediction, due to time constraints.
> >
> > | Backbone | HIV (AUC) $\uparrow$ | Tox21 (AUC) $\uparrow$ | PDBbind (MAE) $\downarrow$ | DUD-E (AUC) $\uparrow$ |
> > |---------|---------|---------|--------|--------|
> > | BIT | **80.2** | **78.1** | **0.927** | **98.4** |
> > | w/o MoDE | 78.3 | 76.9 | 0.977 | 98.0 |
> >
> > Regarding unbalanced data sources, prior work in vision-language pre-training has shown that adding even small amounts of paired image-text data provides value over text-only data, especially for downstream vision-language tasks [6]. Similarly, the inclusion of Q-BioLiP primarily benefits binding-related tasks, with marginal gains for property prediction. This outcome is unsurprising, given the relatively smaller number of molecules available for training in Q-BioLiP, coupled with the fact that PCQM4Mv2 is also utilized by recent baselines (i.e., MoleculeSDE and MoleBLEND).
> >
> > We appreciate these insightful comments. To enhance the model's capabilities, we aim to accumulate more diverse biomolecular training data and scale up model size. This feedback has been invaluable for identifying opportunities and challenges.
> >
> > [6] Su, Weijie, et al. "VL-BERT: Pre-training of Generic Visual-Linguistic Representations." International Conference on Learning Representations. 2020.
> >
> > > There is a lack of a detailed explanation for how positional encodings of 2D and 3D structures are integrated into the model, which leaves a gap in understanding the full architecture and mechanics of the model.
> >
> > We apologize for not providing a detailed description of the positional encodings. We will revise the writing to provide a more comprehensive description. To clarify, the positional encodings for each data format (2D or 3D) comprise two components:
> >
> > - Pairwise encodings: shortest path distance and edge information for 2D graphs [7]; Euclidean distances encoded with the Gaussian Basis Kernel function for 3D structures [8]. They are added to the self-attention module as bias terms.
> >
> > - Atom-wise encodings: degree information for 2D graphs; the sum of distances to all other atoms for 3D structures. They are added to the input atom features.
> >
> > The 2D and 3D positional encodings are integrated into the model through two separate channels. The channel that matches the input format (2D, 3D, or 2D+3D) is activated, while the non-corresponding channel remains disabled.
> >
> > [7] Ying, Chengxuan, et al. "Do transformers really perform badly for graph representation?." Advances in Neural Information Processing Systems 34 (2021): 28877-28888.
> >
> > [8] Shi, Yu, et al. "Benchmarking graphormer on large-scale molecular modeling datasets." arXiv preprint arXiv:2203.04810 (2022).
> >
> > **We hope our response can alleviate your concerns. We are looking forward to further discussion.**

---

> > > ### Comment · Reviewer_2Mm5 · 2023-12-04
> > >
> > > Thank you for your detailed rebuttal and the efforts to address the concerns raised. I appreciate your explanations regarding the use of an invariant architecture, the binding pocket segment consideration, and the integration of positional encodings. These aspects of your paper now have a clearer rationale.
> > >
> > > However, two key concerns remain that I believe require further attention, along with additional concerns shared by other reviewers regarding novelty and technical contribution:
> > > 1. Applicability to a Broader Range of Biomolecules. The assertion that BIT could be a general methodology across various biomolecular contexts is ambitious, and I believe it would benefit from more concrete empirical evidence. Specifically, it would be valuable to see more precise demonstrations of BIT's effectiveness in diverse biomolecular scenarios beyond proteins and small molecules. This would significantly strengthen the claim of BIT being a general-purpose model.
> > > 2. Pre-training Strategy on Different Energy States. The analogy to polysemy in NLP, while interesting, might not fully capture the complexity of pretraining on datasets that span different energy landscapes. The concern here is that the diversity in energy states in your training data might exceed the variability seen in language semantics. Given that the two domain experts in the model are for molecules and proteins, it seems there is a gap in how the model learns the subtleties of different biomolecular contexts. Empirical evidence demonstrating how BIT effectively distinguishes between these diverse contexts would be crucial. It would be beneficial to see further experiments or results that specifically address this aspect.
> > >
> > > Other reviewers, and I concur with them, have raised concerns regarding the novelty and original technical contribution of your work. While the implementation of your approach is commendable, there appears to be a need for a more in-depth analysis and clearer articulation of the novel aspects of your work.

---

### Meta-Review · Area_Chair_zXgJ · 2023-12-05

**Metareview:**

The paper contributes to the quickly growing field of representation learning for molecules. The main challenge in this field is identifying novel and enriching learning signals. In contrast to NLP or vision, there are no natural self-supervised tasks that achieve very significant performance boosts. From this perspective, it is critical to clearly articulate the novel aspects and perform detailed ablations. Such information is necessary for the ICLR and broader community to be able to build on the work and work towards more powerful foundational models for molecules.

The main novelty is in the use of mixture of domain experts, as well as in the overall combination of the specific datasets and architectures. The idea to use protein-ligand complexes along with 3d structures of molecules is not in itself novel.

The strong empirical performance of the model is a major contribution. However, as noted by one of the Reviewers, it is not clear whether the performance gain stems from the improved pretraining objective, the more powerful backbone, or the specific dataset mixture. Given that we are far from transformative foundational models for molecular machine learning, the overall strong performance is not enough to warrant acceptance to ICLR. Let me note here that these criteria differ between venues; I believe the paper would be a great fit for a domain-specific journal (e.g. JCIM) or journals less focused on novelty (e.g. TMLR).

Three out of four reviewers didn't support acceptance of the paper. Two of them have engaged in the rebuttal phase and maintained their scores. The major issues included the aforementioned lack of clarity on comparison, as well as a lack of technical novelty.

All in all, I recommend rejection at this stage. Thank you for the submission, and I hope that the Reviewers’ comments will help improve the paper.

**Justification For Why Not Higher Score:**

See reviewers' key negative comments I described above.

**Justification For Why Not Lower Score:**

N/A

---

### Decision · Program_Chairs · 2024-01-16

Reject